# SEEKING THE SEARCH SPACE FOR SIZE-AWARE VISION TRANSFORMER ARCHITECTURE

## ABSTRACT

Recently, vision transformer methods have gained significant attention due to their superior performance in various tasks, whereas their architectures still highly rely on manual design. Although neural architecture search (NAS) has been introduced to automate the process, it still requires humans to manually specify a fixed search space. Even allowing search space updates, existing methods tend to lose control of model size and result in large and complex models for satisfactory performance. To address these issues, we introduce a constrained optimization framework to **S**eeking the **S**earch **S**pace for **S**ize-aware transformer architecture, named **S4**, which allows the search space to evolve to neighbor search space under user-specified constraints (e.g., model size, FLOPS, etc.). With extensive experiments on various benchmarks, including Cifar10, Cifar100, Tiny ImageNet, and SUN397, the results demonstrate that **S4** can consistently find architectures that align with model size expectations while achieving better performance than those searched by the original search space or with larger size from compared NAS methods. Moreover, we demonstrate the plug-and-play characteristic of **S4** by finding effective yet lightweight adapters for well-recognized foundation models (such as CLIP), achieving excellent performance for downstream tasks.

## 1 INTRODUCTION

Deep learning models often rely heavily on manual design by experts. Neural Architecture Search (NAS) improves the way we search for optimal structures by automating the process. However, the time-consuming and computationally expensive nature of NAS methods poses significant challenges. Thus, one-shot NAS has addressed this issue by training a single supernet encompassing all candidate architectures and efficiently sampling the desired architecture as a subnet from it. The automatically searched architectures using one-shot NAS have demonstrated satisfactory performance in various tasks.

Besides, vision transformer has recently gained great attention in computer vision (CV) due to their remarkable ability to capture long-range dependencies in visual data. Several transformer-based models, including Swin Transformer Liu et al. (2021) and Shunted Transformer Ren et al. (2021) have demonstrated superior performance in CV tasks compared to CNN models. Consequently, there is NAS work like Autoformer (Chen et al., 2021b) that introduced the one-shot NAS technique to search the transformer architecture.

However, conventional one-shot NAS approaches are limited by fixed search space, encountering difficulties in discovering optimal architectures when the search space is unsuitably defined. A recent work S3 (Chen et al., 2021c) has been proposed to address this issue by updating the search space guided by expected and top-tier errors. Nevertheless, a main drawback is that it tends to favor larger architectures to achieve satisfactory performance without taking the model size constraints into consideration during the exploration of the search space. This results in spending extra effort in searching the space that does not meet the model-size constraint, and training and sampling effort on redundant architectures. To tackle the problem, we introduce a constrained optimization framework to size-aware search space exploration for transformer architecture. Our approach ensures the constraints of model size within the search space, eliminating the need for additional filtering in the evolutionary search phase as required by methods like S3. This approach enables the search space

to evolve to better-performed spaces within specified size constraints, facilitating more effective architecture exploration compared to previous methods.

Specifically, we develop a size-aware updating process for search space exploration, which adopts the accuracy gradient ascent to search the overlapped or adjacent search space that satisfies the model size constraints. Namely, we update the search space along the direction which is one of the legitimate directions with the highest cosine similarity to the approximate accuracy gradient. Extensive experiments on diverse benchmarks such as Cifar10, Cifar100, Tiny ImageNet, and SUN397 have demonstrated that **S4** constantly finds models having both superior performance and satisfactory size. Furthermore, our approach shows its plug-and-play characteristic by achieving excellent performance with various base models, including ViT Dosovitskiy et al. (2021), Swin Transformer Liu et al. (2021), Shunted Transformer Ren et al. (2021), and even decoders to adapt the foundation model, CLIP Radford et al. (2021), to the downstream tasks. The primary contributions of our work can be summarized as follows:

- We propose **S4**, a search space exploration strategy for size-aware transformer architecture search, which controls the updating direction with a predefined size range, avoiding extra training and sampling efforts on models that do not align with our size expectations.
- Experiments on Cifar10, Cifar100, Tiny ImageNet, and SUN397 show the effectiveness of **S4** in enhancing search space and finding optimal architectures within the desired size.
- We integrate space-searchable NAS with foundation model and showcase **S4**'s applicability to architecture search by adapting CLIP to other tasks, revealing its versatility.

## 2  RELATED WORK

**Vision transformer**. Transformers are a recent breakthrough in computer vision research, drawing significant attention. In ViT, Dosovitskiy et al. (2021) first replaces traditional Convolutional Neural Network (CNN) with self-attention mechanisms, allowing it to capture long-range dependencies of images. Nevertheless, ViT leads to excessive memory and computational costs dealing with high-resolution images. Wang et al. (2021) introduce a novel approach that employs a hierarchical architecture along with a contracting pyramid to significantly decrease computation on extensive feature maps. In Swin Transformer, Liu et al. (2021) propose a shifted windowing scheme, bringing efficiency by confining self-attention computations to distinct, non-overlapping local windows, while enabling cross-window connections. Yang et al. (2021) propose Focal Transformer that each token attends to nearby tokens for fine-grained details and distant tokens for capturing long-range dependencies. On the other hand, Shunted Transformer Ren et al. (2021) injects heterogeneous receptive field sizes into tokens, handling images with multiple objects in different scales. However, the sparse attention may impose certain restrictions on model's capacity to capture extensive contextual dependencies. To address this limitation, Xia et al. (2022) presents a groundbreaking deformable self-attention module, that adaptively selects key and value pairs' positions in the self-attention process based on data-specific patterns. Chen et al. (2021a) propose the CrossViT framework utilizing separate branches for processing small and large patches and allowing them to complement each other through attention-based fusion modules. Recently, Zhu et al. (2023) proposed BiFormer which utilizes bi-level routing to implement dynamic sparse attention. They initially filter out irrelevant image patches and perform patch-to-patch attention on the remaining patches to reduce computational and memory demands. Without loss of generality, we conduct our experiments on the most representative architectures, ViT Dosovitskiy et al. (2021), Swin Transformer Liu et al. (2021), and Shunted Transformer Ren et al. (2021).

**NAS of one shot and for vision transformer**. We focus on one-shot NAS in this work. The general idea of one-shot NAS is to train a supernet consisting of all architectures in search space to avoid training each subnet independently. Pham et al. (2018) select subnets via a controller module optimized using reinforcement learning. Guo et al. (2020) uniformly sample each path from the supernet for training. Liu et al. (2019) uses a continuous relaxation of the search space to make it differentiable. Wan et al. (2020) propose FBNetV2 to include spatial and channel dimensions in their search approach. Since the booming of ViT, NAS in Transformer has also gained increasing attention. In Autoformer, Chen et al. (2021b) propose a *Weight Entanglement* mechanism to train the supernet. In ViTAS, Su et al. (2021) further proposes a cyclic sampling strategy to improve the training stability. In UniNet, Liu et al. (2022) introduces a unified search space including convolution

block, Transformer block and MLP block. In Pi-NAS, Peng et al. (2021) adopt a non-trivial mean teacher and leverage cross-path learning to reduce feature shifting problems. In PreNAS, Wang et al. (2023) first employs zero-cost proxies to select preferred architectures from isomers, then apply One-shot NAS to achieve better convergence. However, the above methods do not take search space exploration into account, which is an important factor that affects the effectiveness of NAS methods. In NSENet, Ci et al. (2020) propose an evolutionary framework that iteratively updates the search space of CNN-based model, while Chen et al. (2021c) propose the search space searching of Vision Transformer (S3) which evolves the search space of Transformer architecture through a regression approach. Nonetheless, S3 does not control the size of the final architecture. Thus, we propose a novel search space exploration method to seek the optimal architecture given a predefined model size for downstream tasks, which is closer to real application scenarios.

## 3  PROBLEM FORMULATION

Most NAS tasks can be formulated as a constrained optimization problem as follows to search an optimal architecture.

$$
\begin{aligned}
\alpha_{\mathcal{A}}^* &= \arg\min_{\alpha \in \mathcal{A}} \mathcal{L}(W_\alpha^*; \mathcal{D}_{val}) \\
s.t.\ W_\alpha^* &= \arg\min_{W_\alpha} \mathcal{L}(W_\alpha; \mathcal{D}_{train}),\ m(\alpha) \le \mathcal{M},
\end{aligned}
\tag{1}
$$

where $\mathcal{A}$ is a predefined search space, $W_\alpha$ denotes the weights of candidate network architecture $\alpha$, $\mathcal{L}(\cdot)$ is a loss function, $\mathcal{D}_{train}$ and $\mathcal{D}_{val}$ represent train and validation datasets. $m(\cdot)$ is a function calculating the resource consumption of a model and $\mathcal{M}$ is a given resource constraint. Inspired by one-shot NAS (Chen et al., 2021b;c; Su et al., 2021), we solve this constrained optimization problem of Eq. 1 in three stages: (1) exploring the search space satisfying the given constraint, (2) representing the search space by a supernet and optimizing its weights, and (3) searching the optimal architecture from the well-trained supernet by *Evolutionary Search*. Our focus is on the first stage while the second and third stages follow the standard procedures in one-shot NAS for transformers.

The search space $\mathcal{A}$ would encompass all possible architectures, forming an infinite space $\Omega$. However, because of the extensive computational overhead, $\mathcal{A}$ is often a smaller subset of $\Omega$ in practice. To ensure finding a resource-efficient architecture satisfying our constraint, we consider the search space range an important factor. To alleviate the issue of a limited search space, we introduce a size-aware approach for the first stage. Though this paper primarily focuses on model size, our approach can be extended to consider other resource factors such as FLOPs and power consumption.

## 4  SIZE-AWARE SEARCH SPACE EXPLORATION

The ultimate goal of our research is to find the model with the best performance given predefined model size. To achieve this, we first acquire the desired model size range from users, and then establish the upper and lower bounds of each search dimension. This will determine the initial search space w. r. t. the supernet, which is the union of all candidate architectures initially. Instead of a fixed search space, we introduce a constrained optimization framework to update the search space for transformer architectures, which allows the search space to gradually evolve from its neighborhood while adhering to size constraints. The detailed procedure is described as follows.

### 4.1  PRELIMINARY

First, we define $\mathbb{A}^n$ as a set of search spaces composed of $n$ search dimensions. For example, $\mathcal{A} \in \mathbb{A}^4$ implies that $\mathcal{A}$ is one of the search space with defined range for each search dimension in the four-dimensional search space set $\mathbb{A}^4$. $\mathcal{A}$ is represented by a $4 \times 2$ matrix indicating the lower and upper bounds of each dimension in its first and second columns, respectively. Given a search space $\mathcal{A}_i \in R^{4 \times 2}$, the space $\mathcal{A}_j$ is its neighbor space if

$$
\mathcal{A}_j - \mathcal{A}_i =
\begin{bmatrix} \tau_1 \\ \tau_2 \\ \vdots \\ \tau_n \end{bmatrix}
\odot \left(
\begin{bmatrix} \gamma_1^{ij} \\ \gamma_2^{ij} \\ \vdots \\ \gamma_n^{ij} \end{bmatrix}
\begin{bmatrix} 1 & 1 \end{bmatrix} \right),\
\gamma_k^{ij} \in \{-1, 0, 1\},\ \mathcal{A}_i, \mathcal{A}_j \in \mathbb{A}^n,
\tag{2}
$$

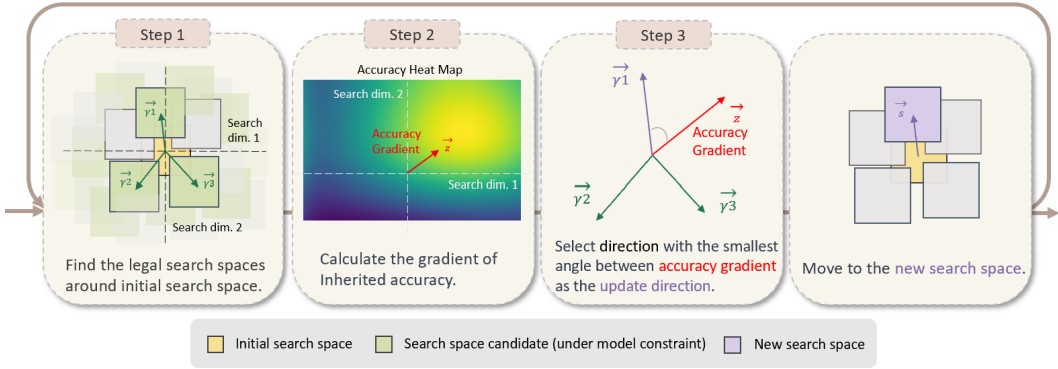

Figure 1: The pipeline of our proposed **S4**. Please refer to Sec. 4 for more details.

where $\tau_k$ is the predefined step size and $\gamma_k^{ij}$ is the step direction of the $k^{th}$ search dimension ($\gamma_k^{ij} = 0$ means remain stationary); $\odot$ denotes element-wise multiplication; $[1 \quad 1]$ is applied to ensure the same increments of the lower- and upper-bound values of each dimension, so as to make the search space associated with the same-scale supernet. The search space $\mathcal{A}_j$ updated within an iteration is at most one step away from $\mathcal{A}_i$ in each search dimension. We use $\mathcal{A}_j \in \mathbb{N}_i$ to express that the search space $\mathcal{A}_j$ is one of the neighbors of $\mathcal{A}_i$. For each $\mathcal{A}_j$ in $\mathbb{N}_i$, the compiled vector $\overrightarrow{\gamma}^{ij} = \begin{bmatrix} \gamma_1^{ij} & \gamma_2^{ij} & \dots & \gamma_n^{ij} \end{bmatrix}$ is known as the relative direction from $\mathcal{A}_i$ to $\mathcal{A}_j$. Step size $\tau_k$ relies only on the $k^{th}$ dimension and remains invariant in the search-space updating process. Its value is set to allow $\mathcal{A}_i$ and $\mathcal{A}_j$ to be immediately adjacent or have a slight overlap, so that there is no gap between the neighbor search spaces.

## 4.2 EXPLORATION PROCEDURE

Our method starts from an initial search space $\mathcal{A}_0$ containing the respective model of target architectures (eg., the original ViT Dosovitskiy et al. (2021)). Assume that we have trained a supernet corresponding to the search space $\mathcal{A}_t$ in the $t^{th}$ iteration. During the search-space evolving, every space (say, $\mathcal{A}_t$) fulfills the desired model-size constraint pre-specified by the users. Thus, the search space is compact; no time will be wasted searching for redundant parts of the search space that violate model size constraints. We leverage the property that the resource consumption (such as model size) often has analytic forms to compute or estimate. Let $m(\cdot)$ denote the model size; eg., consider the target architecture ViT, the model size function w. r. t. the $n = 4$ search dimensions $a, b, c, d$ is

$$m(a, b, c, d) = 1,539d + a(4d + 256cd + 192c + 5d + 7,680 + 2bd^2 + bd + d) + 2d + \#class \times (d+1), \tag{3}$$

with $a, b, c, d$ representing the values of depth, MLP ratio, number of head, and embedding dimension, respectively. $\#class$ is the number of classes when doing classification task in ViT. Similar model size functions can be calculated according to the task currently conducting. Functions of Swin Transformer, Shunted Transformer and CLIP, are given in Appendix C.

The pipeline of search space exploration is shown in Fig. 1. It can be divided into three steps:

***Step 1.*** The first step is to identify legal neighboring search spaces in $\mathbb{N}_t$ for the current search space $\mathcal{A}_t$. Specifically, we limit the size of the largest model in the updated search space to at most $\mathcal{M}$. Only neighboring search spaces that adhere to this constraint are considered as legal candidates. We denote $\mathbb{C}_t$ as the set containing all candidate search spaces (i.e., the legal ones) in $\mathbb{N}_t$,

$$\mathbb{C}_t = \{\mathcal{A}_j | M(\mathcal{A}_j) < \mathcal{M}, \ \mathcal{A}_j \in \mathbb{N}_t\}, \tag{4}$$

where $M(\cdot)$ is a function that calculates the largest model size (i.e, the size corresponding to the model established by the upper-bound values of all search dimensions) in the supernet encoded from the candidate search space. Then, let $\Gamma_t$ be the set of corresponding relative directions from search space $\mathcal{A}_t$ to the candidate search spaces.

***Step 2.*** Secondly, we aim to update to an improved search space that yields enhanced performance in downstream tasks. Let $y = f(\boldsymbol{\beta})$ denote the performance measurement of the model specified by $\boldsymbol{\beta} = (\beta_1, \beta_2, \dots, \beta_n)$, where $\beta_k$ is the value chosen in the $k^{th}$ search dimension and $y$ is the

predicted accuracy of the architecture specified by $\beta$. However, we cannot directly obtain the exact function $f(\cdot)$. Instead, we estimates the optimal gradient-ascent direction $\overrightarrow{z} \approx \nabla f$. First, We sample two architectures $\beta^1$ and $\beta^2$ from the trained supernet, where only the values in the $k^{th}$ search dimension are different and in the other dimensions are the same. Then, we evaluate their performance using the inherited accuracy obtained from the trained supernet. By using the two obtained accuracy values $y^1$ and $y^2$, we calculate the slope of the $k^{th}$ search dimension by computing $slope_k = (\frac{\Delta y}{\Delta \beta})_k = \frac{y^2 - y^1}{\beta_k^2 - \beta_k^1}$. We conduct this sampling and calculating process for each search dimension. Finally, the approximate gradient of accuracy can then be computed as the averaging outcome of $N$ sampled pairs for each search dimension; $N$ is 40 in our experiments.

$$\overrightarrow{z} = \frac{1}{N} \left[ \Sigma_{j=0}^N (\frac{\Delta y_j}{\Delta \beta_j})_1 \quad \Sigma_{j=0}^N (\frac{\Delta y_j}{\Delta \beta_j})_2 \quad \cdots \quad \Sigma_{j=0}^N (\frac{\Delta y_j}{\Delta \beta_j})_n \right] \tag{5}$$

***Step 3.*** After estimating the desired gradient ascent direction $\overrightarrow{z}$, our aim is to find a legal search direction most coherent with it. Thus, we calculate the cosine similarities between $\overrightarrow{z}$ and all the legal search directions $\in \mathbf{\Gamma}_t$, and determine the optimal update direction $\overrightarrow{\gamma}^*$ as the one with the smallest angle to the accuracy gradient $\overrightarrow{z}$ as follows:

$$\overrightarrow{\gamma}^* = \arg\max_{\overrightarrow{\gamma} \in \mathbf{\Gamma}_t} \frac{\overrightarrow{\gamma} \cdot \overrightarrow{z}}{||\overrightarrow{\gamma}|| \cdot ||\overrightarrow{z}||} \tag{6}$$

After evolving the search space with $\overrightarrow{\gamma}^*$, we obtain a new search space not only adheres to our constraints but also improves downstream task performance. When finishing the search space exploration above, we follow the same process as Autoformer (Chen et al., 2021b). We encode the search space into a supernet and optimize it. The process continues until reaching the maximum iterations or when no feasible update directions exist (all cosine similarities are negative). The algorithm detail be referred to Appendix A. Then, we utilize the *Evolutionary Search* algorithm to search for the best architecture within the supernet. Finally, we perform a retraining process from scratch. This allows us to obtain the best architecture given the predefined size constraints.

## 5 DISCUSSION

We discuss and compare our proposed method with S3 (Chen et al., 2021c), another search space exploration approach. While S3 estimates the updating direction by fitting a linear function for each search dimension based on the expected and top-tier error of sampled architectures, it does not impose model size constraints in the search space. Hence, the method does not ensure whether the supernet required to be trained in the current step has legal solutions or not. Instead, S3 leaves the model size constraint to be addressed after supernet training, and then discard any sampled architecture exceeding this constraint. Therefore, both the training and sampling efforts spent on unqualified and redundant parts of the supernet are wasteful. The un-controlled resource consumption in the search space of S3 makes it difficult to find solutions fulfilling the model size constraint. We always need several times of trial-and-error to get a satisfied solution.

Furthermore, unlike our approach that moves a fixed step-size to the neighbor space, the movement control within the search space in S3 is irregular and requires extensive human labor of tuning. In the experiments, we have shown the performance comparison with S3. However, it requires extensive offline tuning to obtain the results, making it difficult to use in practice.

Our S4 guarantees that the model size stays within the predefined range by controlling the exploration direction of the search space in the first stage. Since our approach has *constrained the model size in the search-space level*, we ensure all models sampled in the Evolutionary Search stage being qualified, making it able to fully employ the power of Evolution Search without constraints. Our approach uses a regular movement step for each dimension, which can easily control that no gap exists between the current and new search spaces and a consistent rate of neighbor-space overlapping is applied. Conversely, the uncontrollable model size and the irregular step-size movement of S3 makes it inconvenient and contradict to the principle of NAS, which aims to eliminate manual involvement. Our method enables exploring an improved search space within the given model constraint, avoiding the need for extensive tuning, which offers better convenience in practical applications.

Table 1: Initial search space of ViT, Swin Transformer, Shunted Transformer and CLIP.

| Models | Embed Dim | Depth Num | Head Num | MLP Ratio | Params Range |
|---|---|---|---|---|---|
| ViT | (336, 384, 24) | (12, 14, 1) | (2, 14, 2) | (3.5, 4.0, 0.5) | 23M~35M |
| Swin | (72, 96, 12) | (10, 14, 2) | - | (3.5, 4.0, 0.5) | 14M~35M |
| Shunted | (36, 60, 12) | (15, 19, 4) | - | (6, 8, 1) | 5M~21M |
| CLIP | (336, 384, 24) | (3, 4, 1) | (2, 14, 2) | (3.5, 4.0, 0.5) | 16M~35M |

## 6 EXPERIMENTS

To validate the effectiveness of our method, we apply it to different Vision Transformer architectures for evaluation, including ViT Dosovitskiy et al. (2021), Swin Transformer Liu et al. (2021), and Shunted Transformer Ren et al. (2021). Moreover, we integrate our approach with foundational models like CLIP (Radford et al., 2021), where we search for decoders architecture to complement the fixed encoder of CLIP to downstream tasks.

### 6.1 IMPLEMENTATION DETAILS

***Space Searching.*** The initial search space is given in Tab. 1. **ViT.** For ViT, the search space is defined as number of blocks, embedding dimension, number of heads and MLP ratio (the expansion ratio of the hidden layer dimension to the embedding dimension). The number of blocks and embedding dimension are set to $\{12, 13, 14\}$ and $\{336, 360, 384\}$, respectively, and the number of heads and MLP ratio are set to $\{2, 4, 6, 8, 10, 12, 14\}$ and $\{3.5, 4.0\}$, respectively. Note that the number of heads and MLP ratio can be different between blocks.
**Swin Transformer and Shunted Transformer.** Different from ViT, Swin and Shunted Transformers possess hierarchical architecture (multiple stages) as illustrated in Fig. 2. In each stage, it may contain different numbers of encoder block. We design the search dimensions to be the number of blocks in each stage, MLP ratio and embedding dimension, with the number of heads remaining constant. In Swin Tranformer, We adopt 3, 6, 12 and 12 heads for the blocks in 4 stages respectively and they are fixed. The number of blocks in each stage is set as 2, 2, $x$, 2, respectively, where $x$ is one of the search dimensions that we will explore, which is set as $\{4, 6, 8\}$. The embedding dimension and MLP ratio are $\{72, 84, 96\}$ and $\{3.0, 3.5, 4.0\}$ respectively. In Shunted Transformer, the number of stages is also 4, and we set the number of blocks in each stage to 2, $x$, $3x$, 1, respectively, where $x$ is set as $\{3, 4\}$. The number of heads are fixed and set as 2, 4, 8, 16 associated with the 4 stages. The embedding dimension and MLP ratio are set to $\{36, 48, 60\}$ and $\{6, 7, 8\}$ respectively.
**CLIP.** Recently, the decoder architecture introduced in LST (Sung et al., 2022) and EVL (Lin et al., 2022) allows user to efficiently adapt the popular foundation model, CLIP Radford et al. (2021), to downstream tasks with superior performance when resources are limited. We further explore the search dimensions of the decoder design for better performance and resource usage, including the number of decoder blocks, the number of heads, embedding dimension, and the MLP ratio as illustrated in Fig. 3. The model size constraint is set as 30M. In addition, the search dimensions are defined as $\{3.5, 4.0\}$ for MLP ratio, $\{2, 4, 6, 8, 10, 12, 14\}$ for number of heads, $\{3, 4\}$ for number of decoder blocks and $\{336, 360, 24\}$ for embedding dimension.
All the model-size constraint are set according to the original model (30M for ViT, 30M for Swin and 22M for Shunted Transformer, as the CLIP-based decoders constraint is referred to EVL (Lin et al., 2022) and set to 30M.) More details can be found in Appendix B. Let $T$ to be the maximum number of iterations of search space exploration, and we set $T = 3$ in our experiments, which follows the setting in S3. Every supernet is trained for 500 epochs.

***Supernet training.*** We train the supernets using the technique referred to AutoFormer (Chen et al., 2021b). Data augmentation techniques such as random erasing, Cutmix (Yun et al., 2019), Mixup (Zhang et al., 2018) and RandAugment (Cubuk et al., 2019) are used for training. AdamW (Loshchilov & Hutter, 2017) is adopted as our optimizer, where the initial and minimum learning rates are set as $5 \times 10^{-4}$ and $1 \times 10^{-5}$, respectively in a cosine scheduler with 5-epoch warmup. The weight decay, batch size, label smoothing, and drop rate for stochastic depth are set to 0.05, 128, 0.1, and 0.2, respectively.

***Model Retraining.*** After *Evolutionary Search*, the models are retrained for 300 epochs, and the other settings are all same as supernet training.

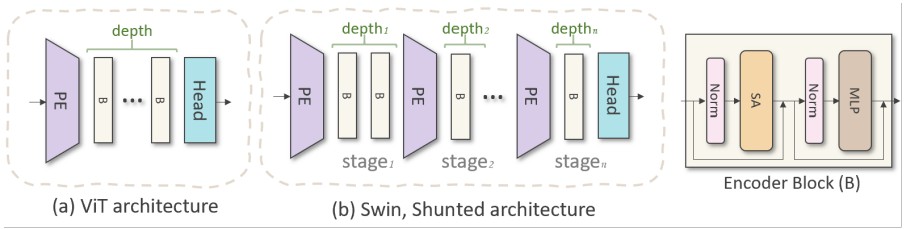

Figure 2: The architecture of ViT, Swin Transformer, and Shunted Transformer. (PE: Patch Embedding, SA: Self-attention, MLP: Multi-layer perceptron.)

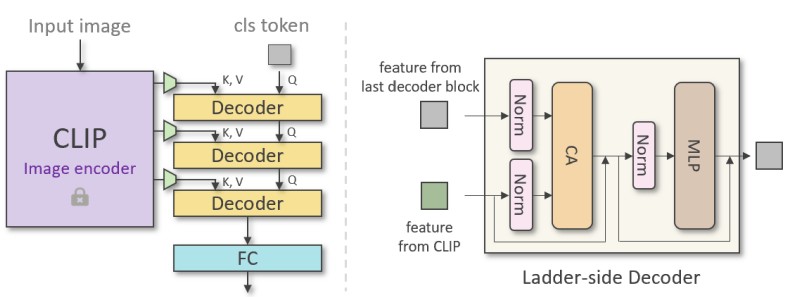

Figure 3: The architecture of CLIP with a Ladder-side decoder (Sung et al., 2022; Lin et al., 2022).

## 6.2 RESULTS ON IMAGE CLASSIFICATION

**Vision Transformer** Tab. 2 shows the performance of different NAS methods searching ViT architecture for the Cifar10, Cifar100, Tiny ImageNet, and SUN397 datasets. Note that the former two datasets are the representative datasets in image classification tasks and the latter two are large datasets having $100,000$ images of 200 classes and 100,000 images of 397 classes, respectively. Since the original papers did not include the results on above datasets, we reproduced the compared methods ourselves. All the experiments are conducted by training from scratch without using any pretrained model, following similar setting as Chen et al. (2021b) except the resolutions of input images are scaled to $224 \times 224$ instead of $384 \times 384$ for the purpose of quick proof of concept. We compare our method with pure ViT and Transformer-based NAS methods, Autoformer and S3.

It can be observed that by using the proposed **S4** the performance can be boosted on all datasets. Moreover, the model sizes of the architectures found by our method (ViT + **S4**) is smaller compared to other methods. Further experiments about the comparison of S3 and our method will be shown in Sec. 7. To sum up, the results of Tab. 2 reveal that our method can find highly performing architectures with constrained model size.

**Swin Transformer, Shunted Transformer and CLIP-based model** Tab. 3 shows the performance of Swin Transformer, Shunted Transformer and the CLIP-based model on the four datasets (Cifar10, Cifar100, Tiny ImageNet and SUN397). As can be seen, our **S4** method is also effective on searching better Swin Transformer, Shunted Transformer architecture and adaptive decoders for CLIP, where all the performances are improved with our design. On the other hand, our approach consistently identifies relatively lightweight models in each case, demonstrating its ability to discover effective and lightweight models. In summary, the results highlight the generality and effectiveness of our method on size-constrained exploration of the search space for NAS when leveraging both transformer variants and foundation model.

## 7 EXPERIMENTAL ANALYSIS BETWEEN S3 AND **S4**

**Detailed analysis of search iterations** Here, we delve into the detailed performance and model size obtained from each search iteration and conduct comparative analyses between S3 and our method.

Table 2: Top-1 Accuracy of **S4** on Cifar10, Cifar100, Tiny ImageNet, and SUN397 compared with other methods searching Vision Transformer architecture. For S3 and **S4**, we set the model size constraint as 30M for the fair comparison purpose with the original ViT and AutoFormer-S. (We train each NAS method using Cifar10, Cifar100 and evaluate them with images scaled to $224 \times 224$ input resolution.)

| Models | #Search Space Update | Backtrack | #Params.(M) | Cifar10 | Cifar100 | Tiny ImageNet | SUN397 |
|---|---|---|---|---|---|---|---|
| ViT | - | | 29/29/29/30 | 93.59 | 76.45 | 67.16 | 68.33 |
| ViT + AutoFormer-S | 0 | - | 28/28/30/30 | 93.89 | 77.83 | 67.43 | 68.54 |
| ViT + S3 | 3 | ✓ | 29/29/27/27 | 94.23 | 78.02 | 67.36 | 68.42 |
| **ViT + S4 (ours)** | 3 | ✗ | 25/26/27/29 | **94.32** | **78.68** | **67.92** | **68.82** |

Table 3: Comparison of Transformer variants and CLIP-based architecture with or without **S4** on Cifar10, Cifar100, Tiny ImageNet and SUN397.

| Models | #Params.(M) | Cifar10 | Cifar100 | Tiny ImageNet | SUN397 | searched block |
|---|---|---|---|---|---|---|
| Swin (Liu et al., 2021) | 27/28/28/28/28 | 94.39 | 79.03 | 71.9 | 73.55 | - |
| **Swin+S4 (ours)** | 25/26/28/28 | 95.65 | 80.31 | 73.09 | 74.32 | encoder |
| Shunted (Ren et al., 2021) | 22/22/22/23/23 | 97.39 | 84.36 | 74.07 | 76.03 | - |
| **Shunted+S4 (ours)** | 20/21/21/21 | **97.67** | 85.66 | 75.59 | 78.19 | encoder |
| CLIP (Radford et al., 2021)+LSD | 31/31/31/32/32 | 97.21 | 86.44 | 79.81 | 83.35 | - |
| **CLIP+LSD+S4 (ours)** | 20/26/28/30 | 97.23 | **87.63** | **81.35** | **85.87** | decoder |

We have applied both methods to ViT and the results are presented in Tab. 4 for Tiny ImageNet, and more results can be found in Appendix D. As indicated by Tab. 4 , we observe that S3 struggled to discover models that adhere to the model-size constraints after several iterations of search space updates. This issue persisted despite our repeated attempts to tune the quantization hyperparameter in S3. Note that tuning the hyperparameter posed significant challenges. When the hyperparamter increased, the quantized factor often resulted in a zero value due to the floor function, leading to a stable search space that did not update. Conversely, decreasing the hyperparameter value could lead to dramatic changes in the search space since it serves as the denominator of the slope term. Consequently, model sizes could easily grow excessively and become uncontrollable. In short, the hyperparameter of S3 needs to be carefully adjusted and is specific for each type of models and datasets, which is labor-intensive and violate the main idea of NAS, aiming to reduce the manual design efforts. Conversely, due to relatively slow and controlled updating process inherited from our method, the proposed method is able to gradually evolve to a better search space. In addition, the model with appropriate size can be consistently found and achieves superior performance as the number of search iterations increases.

Since S3 failed to find models adhering the model size constraint, making the discovery of qualified models in the Evolutionary stage impossible, we have adjusted the searching procedure for S3, in order to conduct further comparisons with our method on the same level. We retroactively backtrack to the search space found in the $(T-1)^{th}$ update iteration and verify whether any feasible solutions exist by using our derived model size function. If we still fail to find the solution in the $(T-1)^{th}$ search space, we then go back to the $(T-2)^{th}$ search space, and so on. The performance is shown in Tab. 5 on Tiny ImageNet and SUN397 (tables for Cifar10 and Cifar100 are in Appendix D). Take the results on Tiny ImageNet for example, we needed to backtrack the S3 method twice to find the suitable models for ViT, Swin and Shunted Transformers. Even with the backtracking, our method still consistently outperforms S3, with accuracy improvements of 0.66, 0.79, and 0.88, while retaining smaller model size in most cases.

Table 4: Comparison of S3 and **S4** under model constraint in each exploration iteration on Tiny ImageNet. We report the retrained accuracy of architectures sampled from the $t^{th}$ search space.

| Model | Exploring approach | Search iteration | #Params.(M) | Tiny ImageNet Accuracy |
|---|---|---|---|---|
| ViT (Dosovitskiy et al., 2021) | S3 | 1 | 27.88 | 67.36 |
| ViT | S3 | 2 | - | - |
| ViT | S3 | 3 | - | - |
| ViT | **S4 (ours)** | 1 | 26.14 | 67.39 |
| ViT | **S4 (ours)** | 2 | 26.19 | 67.55 |
| ViT | **S4 (ours)** | 3 | 27.45 | **67.92** |

Table 5: Comparisons of search space exploration of ViT, Swin Transformer, and Shunted Transformer using S3 and **S4**. It is worth noting that S3 could not find any architecture satisfying the specified constraints after search space evolution and require to backtrack to the previous search space for accomplishing the architecture search. In contrast, the proposed **S4** finds a better performing architecture under the constraint without carefully tuning steps.

| Model | Exploring approach | backtrack time | Model Constraint(M) | #Params.(M) | Tiny ImageNet | SUN397 |
|-------|-------------------|----------------|---------------------|-------------|---------------|--------|
| ViT | - | - | 30 | 28.91/29.56 | 67.16 | 68.33 |
| ViT | S3 | 2/2 | 30 | 27.88/27.39 | 67.36 | 68.42 |
| ViT | **S4 (ours)** | 0/0 | 30 | 27.45/28.73 | **67.92 (+0.56)** | **68.82(+0.4)** |
| Swin | - | - | 30 | 27.67/27.82 | 71.9 | 73.55 |
| Swin | S3 | 2/1 | 30 | 27.58/29.75 | 72.4 | 74.05 |
| Swin | **S4 (ours)** | 0/0 | 30 | 28.43/28.43 | **73.09 (+0.69)** | **74.32(+0.27)** |
| Shunted | - | - | 22 | 21.99/22.09 | 74.07 | 76.03 |
| Shunted | S3 | 2/2 | 22 | 21.32/21.38 | 75.16 | 77.64 |
| Shunted | **S4 (ours)** | 0/0 | 22 | 21.27/21.47 | **75.59 (+0.43)** | **78.19(+0.55)** |

Table 6: Comparisons of search space exploration of ViT, Swin Transformer, and Shunted Transformer using S3 and **S4** in the scenario without model size constraints. It is worth noting that under the same unconstrained conditions, the proposed method can find not only better performing but also more lightweight architectures than S3. Moreover, our constrained version can also find much more compact architecture with better or comparable performances than S3.

| Model | Exploring approach | w/ constraint | #Params.(M)(Cifar10/Cifar100) | Cifar10 | Cifar100 |
|-------|-------------------|---------------|-------------------------------|---------|----------|
| ViT (Dosovitskiy et al., 2021) | S3 (un-constrained) | | 44.65/44.88 | 94.35 | 78.24 |
| ViT | **S4 (ours)** | ✓ | 25.39/26.45 | 94.32 | 78.68 |
| ViT | **S4 (un-constrained) (ours)** | | 35.41/38.72 | **94.49** | **79.11** |
| Swin (Liu et al., 2021) | S3 (un-constrained) | | 52.81/56.2 | 95.91 | 80.86 |
| Swin | **S4 (ours)** | ✓ | 25.37/26.24 | 95.65 | 80.31 |
| Swin | **S4 (un-constrained) (ours)** | | 37.45/38.2 | **96.03** | **81.15** |
| Shunted (Ren et al., 2021) | S3 (un-constrained) | | 37.44/42.2 | 97.82 | 85.83 |
| Shunted | **S4 (ours)** | ✓ | 20.18/21.56 | 97.67 | 85.66 |
| Shunted | **S4 (un-constrained) (ours)** | | 33.38/34.94 | **97.85** | **85.92** |

Based on the experiments conducted, we conclude that our method possesses a plug-and-play characteristic, exhibiting compatibility with various transformer-based models. Furthermore, it consistently demonstrates superior performance across several benchmarks, including large datasets with hundreds of classes. This reaffirms the effectiveness of our approach in identifying excellent models within the specified size constraints.

**Unconstrained Scenario.** We are also interested in how our updating design performs when there is no model size constraints. Thus, we conducted an experiment that enables the search space to evolve in a favorable direction without imposing any model size constraints. The results are shown in Tab. 6. For the unconstrained **S4**, we outperform S3 0.14%/0.12%/0.03% on Cifar10 for ViT, Swin and Shunted Transformers, respectively. Similarly, we achieve 0.87%/0.29%/0.09% improvements on Cifar100. It's worth noting that all the unconstrained models we searched for are smaller than those obtained from unconstrained S3. From this observation, it can be inferred that even without an explicit size constraint, our neighboring updating design makes the model gradually update to a better search space, rather than dramatically changes to a new search space with a large size. Consequently, our approach results in a smaller architecture sampled then the S3 method.

## 8 CONCLUSION

In this paper, we introduce a constrained optimization framework for transformer architecture search with search space exploration. Our approach combines accuracy gradient ascent with discrete neighboring search space evolution while adhering to strict model size constraints. Unlike previous methods, our approach ensures the discovery of the optimal architecture within a predefined model size range. Extensive experiments demonstrate its plug-and-play characteristic and superior performance. We aim to extend its applicability to a wider range of tasks (object detection, semantic segmentation) and model backbones (CNN-based) as our future work.

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
