## Supplementary Materials

This supplementary material contains additional details of Sec. 4, 6.1 and 7, and illustrations of the effectiveness for the proposed method.

## A  SIZE-AWARE SEARCH SPACE EXPLORATION

Besides the descriptions in Sec. 4, we also provide the full pseudo code of the proposed method for search space evolution (i.e., step 1 to step 3) during architecture search in Algorithm 1.

---

**Algorithm 1** Size-Aware Search Space Exploration

---

**Input:** Infinite search space, $\Omega$; Max number of the iterations, $T$; Number of sampling times, $N$; Model Constraint, $\mathcal{M}$; Search dimensions in search space, $\beta_1, \ldots, \beta_n$; Candidate choices in the $k^{th}$ search dimension, $\mathbb{V}_k$; Evolving steps for each search dimension, $\tau_1, \ldots, \tau_n$;

**Output:** The most promising search space $\mathcal{A}^*$

1: Initialize a search space $\mathcal{A}_0$ from $\Omega$
2: **for** $t = 1, 2, \ldots, T$ **do**
3:     Get the set of legal search space candidates $\mathbb{C} = \{\mathcal{A}_j | \mathcal{A}_j - \mathcal{A}_{t-1} = \begin{bmatrix} \gamma_1^{t-1,j}\tau_1 & \gamma_2^{t-1,j}\tau_2 & \ldots & \gamma_n^{t-1,j}\tau_n \end{bmatrix}, M(\mathcal{A}) \leq \mathcal{M}\}$, where $\gamma_k^{t-1,j} \in \{-1, 0, 1\}$.
4:     Optimize the weights $W_{\mathcal{A}_t}$ of supernet corresponding to the space $\mathcal{A}_t$
5:     **for** $k = 1, 2, \ldots, n$ **do**
6:         **for** $i = 1, 2, \ldots, N$ **do**
7:             Randomly sample an architecture $\alpha$ from the trained supernet.
8:             Get the set $\mathbb{S}$ of architectures according to $\beta^{cand}$ where $\beta_k^{cand} \in \mathbb{V}_k$ and $\beta_k^{cand} \neq \beta_k^\alpha$, while the other dimensions are the same as $\alpha$.
9:             Get the inherited accuracy $y$ of each architecture in $\mathbb{S}$.
10:            Calculate the $slope_k = (\frac{\Delta y}{\Delta \beta})_k = \frac{y^2 - y^1}{\beta_k^2 - \beta_k^1}$ for each pair $\beta^1$, $\beta^2$ in $\mathbb{S}$.
11:        **end for**
12:    **end for**
13:    Get the final accuracy gradient $\overrightarrow{z}$ according to Eq. 5
14:    Find the update direction $\overrightarrow{\gamma}^*$ according to Eq. 6
15:    **if** $\frac{\overrightarrow{z} \cdot \overrightarrow{\gamma}^*}{||\overrightarrow{z}|| \cdot ||\overrightarrow{\gamma}^*||} \leq 0$ **then**
16:        break
17:    **end if**
18:    $\mathcal{A}^* = \mathcal{A}_{t-1} + \begin{bmatrix} \gamma_1^*\tau_1 & \gamma_2^*\tau_2 & \ldots & \gamma_n^*\tau_n \end{bmatrix}$
19:    $\mathcal{A}_t = \mathcal{A}^*$
20: **end for**

---

## B  MORE DETAILS OF SEARCH SPACE

### B.1  DISCUSSION OF SEARCH SPACE DESIGN

**Swin Transformer and Shunted Transformer.** We observe that the released Swin-T, Swin-S, Swin-B and Swin-L models all having 2 for the depth of the 1st, 2nd and 4th stages. Thus we design the initial search space of Swin Transformer with only the depth of the 3rd stage exploring. Similarly, the released Shunted-T, Shunted-S, and Shunted-B models feature a depth of 2 for the 1st and 4th stages, while the depth of the 3rd stage is three times that of the 2nd stage. Thus we explore the depth of the 2nd and 3rd stages following this pattern.

**CLIP.** The decoder blocks in our experiment is served as the adapters of CLIP, allowing users to leverage foundational models without extensive fine-tuning, especially when resources are limited. Thus, finding constrained-size models while remaining excellent performance is vital, which is the essence of this paper.

## B.2 RELATIONSHIP BETWEEN NEIGHBORS

The detailed relationship between a search space and its neighbors is shown in Fig. 4. The neighboring search space can be at most one step away from the original search space in each search dimension. The search dimensions are discrete, and in practice, we search for architectures with integer values for them.

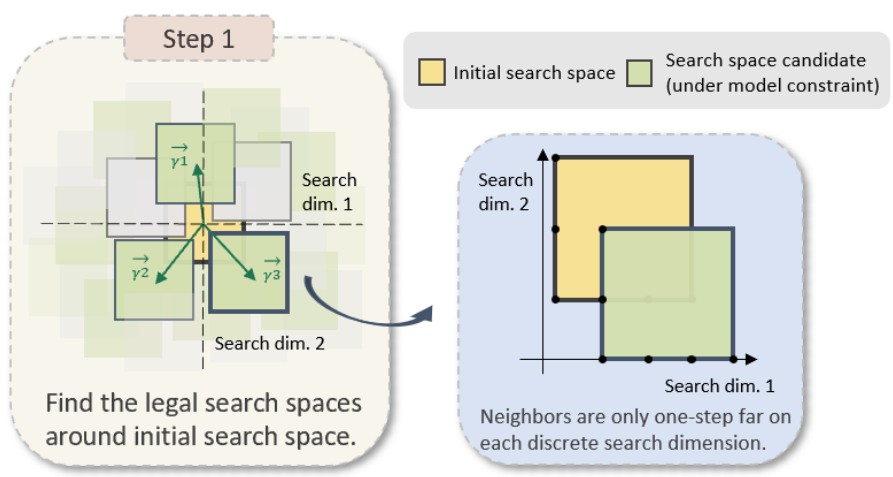

Figure 4: Detailed illustration of the relationship between initial search space and its neighbors. Neighbor search space can be at most one step away from the original search space in each discrete search dimension.

## C MODEL SIZE FUNCTION

Vision Transformer Dosovitskiy et al. (2021) has the same embedding dimension in every block, while Swin Transformer Liu et al. (2021) and Shunted Transformer Ren et al. (2021) have different embedding dimensions in different stages. There is a Patch Merging module between the stages, and the dimension gets doubled. The architectures are illustrated in Figure 2 in the main paper. Assume that the value of depth, MLP ratio, number of head, and embedding dimension are $a, b, c, d$, respectively.

**Vision Transformer.** Before the $1^{st}$ block, there is a patch embedding module, which transforms an input image into a feature embedding, implemented by a CNN layer, where the number of parameters is $d \times (16 \times 16 \times 3) + d$, $16 \times 16$ is a given patch size, and 3 is the number of input channels. Two parameters, class token and positional embedding, are also calculated, which are $d, d \times (\#patches + 1)$, respectively. In each block, it consists of multiple modules, two Layer-Norms, an attention module, and an MLP module. The number of parameters in aforementioned modules are $4d, 256cd + 192c + 5d + 7680, 2bd^2 + bd + d$, respectively. At last, there are a Layer-Norm module and a prediction head, which includes $2d$ and $\#classes \times (d + 1)$ parameters. The overall number of parameters is thus

$$m_{vit}(a, b, c, d) = 1,539d + a(4d + 256cd + 192c + 5d + 7,680 + 2bd^2 + bd + d) + 2d + \#class \times (d+1), \quad (7)$$

where $\#class$ is the number of classes when applying ViT to a classification problem.

**Swin Transformer.** We set the patch size to 4, window size to 7, number of heads to 3, 6, 12, 12 in 4 stages, respectively. Before the $1^{st}$ stage, there is also a patch embedding module, implemented by a CNN layer, where the number of parameters is $d \times (4 \times 4 \times 3) + d$ with $4 \times 4$ a given patch size, and 3 the number of input channel, respectively. Swin Transformer does not have class tokens. In each block, there is an extra parameter, relative positional bias, which contains $13 \times 13 \times \#heads$ parameters, where 13 comes from $2 \times window\_size - 1$, and each block consists of multiple modules, two LayerNorm, an attention module, and an MLP module. The number of

parameters in aforementioned modules are $2 \times 2d, 4d^2 + 4d, 2bd^2 + bd + d$, respectively. Thus, the number of parameter is $169c + 2bd^2 + 4d^2 + bd + 9d$ in one block. After each stage except the last one, there is a patch merging module and a LayerNorm layer, whose number of parameter are $8d^2, 8d$, respectively. At last, there are a LayerNorm module and a prediction head, which include $2d_4$ and $\#classes \times (d_4 + 1)$ parameters, respectively. Therefore, the overall number of parameters is given in Eq. 8.

$$
\begin{aligned}
m_{swin}(a, b, d) = {} & 49d_1 + 2(169c_1 + 2bd_1^2 + 4d_1^2 + bd_1 + 9d_1) + 8d_1^2 + 8d_1 \\
& + 2(169c_2 + 2bd_2^2 + 4d_2^2 + bd_2 + 9d_2) + 8d_2^2 + 8d_2 \\
& + a(169c_3 + 2bd_3^2 + 4d_3^2 + bd_3 + 9d_3) + 8d_3^2 + 8d_3 \\
& + 2(169c_4 + 2bd_4^2 + 4d_4^2 + bd_4 + 9d_4) + 2d_4 + \#classes \times (d_4 + 1),
\end{aligned}
\tag{8}
$$

where $\#class$ is the number of classes when applying Swin Transformer to a classification problem, and $[c_1 \quad c_2 \quad c_3 \quad c_4] = [3 \quad 6 \quad 12 \quad 12], [d_1 \quad d_2 \quad d_3 \quad d_4] = [d \quad 2d \quad 4d \quad 8d]$.

**Shunted Transformer.** We set the window size to 7, number of heads to 2, 4, 8, 16 in 4 stages, respectively. Before the $1^{st}$ stage, there is a patch embedding module, implemented by two CNN layer with the respective number of parameters $3 \times 3 \times 64 \times 3, 2 \times 2 \times 64 \times d + d$, and a LayerNorm with the number of parameters $2d$. Shunted Transformer does not have class tokens and positional embedding. In each block, two LayerNorms, an attention module and an MLP module are included, which have the sizes $2 \times 2d, 4d^2 + 8d, 2bd^2 + bd + d$, respectively. After each stage, there is a LayerNorm layer as well, whose number of parameters is $2d$. Between two stage, there are a patch merging module, making the embedding dimension doubled, and LayerNorm layer, containing $3 \times 3 \times d_i \times d_{i+1}, 2 \times d_{i+1}$. At last, there is a prediction head, which includes $\#classes \times (d_4 + 1)$ parameters. Therefore, the overall number of parameters is shown in Eq. 9.

$$
\begin{aligned}
m_{shunt}(a, b, d) = {} & 1728 + 259d_1 + 2(2bd_1^2 + 4d_1^2 + bd_1 + 13d_1) + 2d_1 \\
& + 9d_1 \times d_2 + 2d_2 + a(2bd_2^2 + 4d_2^2 + bd_2 + 13d_2) + 2d_2 \\
& + 9d_2 \times d_3 + 2d_3 + 3a(2bd_3^2 + 4d_3^2 + bd_3 + 13d_3) + 2d_3 \\
& + 9d_3 \times d_4 + 2d_4 + (2bd_4^2 + 4d_4^2 + bd_4 + 13d_4) + 2d_4 + \#classes \times (d_4 + 1),
\end{aligned}
\tag{9}
$$

where $\#class$ is the number of classes when applying Swin Transformer to a classification problem, and $[c_1 \quad c_2 \quad c_3 \quad c_4] = [2 \quad 4 \quad 8 \quad 16], [d_1 \quad d_2 \quad d_3 \quad d_4] = [d \quad 2d \quad 4d \quad 8d]$.

**CLIP.** The Ladder-side decoder is composed of a cross-attention module and a MLP module with LayerNorms. The cross attention module can be calculated as $8d^2 + 7d$ and the MLP module can be calculated as $bd^2$. LayerNorms is calculated as $6d$. At last, there are another LayerNorm module and a prediction head, which includes $2d$ and $\#classes \times (d + 1)$ parameters, respectively. Therefore, the overall number of parameters is given in Eq. 10.

$$
\begin{aligned}
m_{CLIP}(a, b, d) = {} & 24d + b_1 d_1^2 + 8d_1^2 + 7d_1 + 6d_1 \\
& + b_2 d_2^2 + 8d_2^2 + 7d_2 + 6d_2 \\
& + b_3 d_3^2 + 8d_1^3 + 7d_3 + 6d_3 \\
& + b_4 d_1^4 + 8d_1^4 + 7d_4 + 6d_4 + 2d_4 + \#classes \times (d_4 + 1),
\end{aligned}
\tag{10}
$$

Table 7: Comparison of S3 and **S4** under model constraint in each exploration iteration on Cifar10. We report the retrained accuracy of architectures sampled from the $t^{th}$ search space.

| Model | Exploring approach | Search iteration | #Params.(M) | Cifar 10 Accuracy |
|---|---|---|---|---|
| ViT (Dosovitskiy et al., 2021) | S3 | 1 | 28.78 | 94.17 |
| ViT | S3 | 2 | 29.32 | 94.23 |
| ViT | S3 | 3 | - | - |
| ViT | **S4 (ours)** | 1 | 23.17 | 94.09 |
| ViT | **S4 (ours)** | 2 | 25.25 | 94.25 |
| ViT | **S4 (ours)** | 3 | 25.39 | **94.32** |
| Swin (Liu et al., 2021) | S3 | 1 | 29.88 | 95.61 |
| Swin | S3 | 2 | - | - |
| Swin | S3 | 3 | - | - |
| Swin | **S4 (ours)** | 1 | 27.09 | 95.55 |
| Swin | **S4 (ours)** | 2 | 27.7 | 95.63 |
| Swin | **S4 (ours)** | 3 | 28.33 | **95.65** |
| Shunted (Ren et al., 2021) | S3 | 1 | 21.28 | 97.6 |
| Shunted | S3 | 2 | - | - |
| Shunted | S3 | 3 | - | - |
| Shunted | **S4 (ours)** | 1 | 20.3 | 97.58 |
| Shunted | **S4 (ours)** | 2 | 21.98 | **97.67** |

Table 8: Comparison of S3 and **S4** under model constraint in each exploration iteration on Cifar100. We report the retrained accuracy of architectures sampled from the $t^{th}$ search space.

| Model | Exploring approach | Search iteration | #Params.(M) | Cifar 100 Accuracy |
|---|---|---|---|---|
| ViT (Dosovitskiy et al., 2021) | S3 | 1 | 29.1 | 78.02 |
| ViT | S3 | 2 | - | - |
| ViT | S3 | 3 | - | - |
| ViT | **S4 (ours)** | 1 | 24.32 | 78.11 |
| ViT | **S4 (ours)** | 2 | 26.44 | 78.52 |
| ViT | **S4 (ours)** | 3 | 26.45 | **78.68** |
| Swin (Liu et al., 2021) | S3 | 1 | 28.47 | 79.52 |
| Swin | S3 | 2 | - | - |
| Swin | S3 | 3 | - | - |
| Swin | **S4 (ours)** | 1 | 25.6 | 79.63 |
| Swin | **S4 (ours)** | 2 | 25.56 | 79.98 |
| Swin | **S4 (ours)** | 3 | 26.24 | **80.31** |
| Shunted (Ren et al., 2021) | S3 | 1 | 19.4 | 84.78 |
| Shunted | S3 | 2 | - | - |
| Shunted | S3 | 3 | - | - |
| Shunted | **S4 (ours)** | 1 | 20.16 | 84.8 |
| Shunted | **S4 (ours)** | 2 | 21.7 | 85.57 |
| Shunted | **S4 (ours)** | 3 | 21.56 | **85.66** |

# D   MORE EXPERIMENTAL RESULTS

## D.1   RESULTS IN SEARCH ITERATIONS

In the main paper, we provide the comparison of S3 and S4 in each exploration iteration on Tiny ImageNet for ViT in Fig. 4. Here we list the complete results for ViT, Swin Transformer and Shunted Transformer on Cifar10, Cifar100 and Tiny ImageNet in Tab. 7, Tab. 8 and Tab. 9, respectively. Similar results can be seen that our method consistently outperforms S3 after three iterations of search space exploration, while S3 fails to discover models adhering to the size constraint.

## D.2   RESULTS WITH BACKTRACKING

In the main paper, we provide the comparison of S3 and S4 on Tiny ImageNet and SUN397, with backtracking applied on S3 to meet the model constraints. We further list the result on Cifar10 and Cifar100 in Tab. 10 and Tab. 11. Results are consistent with the performance on Tiny ImageNet and SUN397. Though S3 could find models under the size limit with backtracking, they are suboptimal.

Table 9: Comparisons of S3 and **S4** under model constraint in each exploration iteration on Tiny ImageNet. We report the retrained accuracy of architectures sampled from the $t^{th}$ search space.

| Model | Exploring approach | Search iteration | #Params.(M) | Tiny ImageNet Accuracy |
|---|---|---|---|---|
| ViT (Dosovitskiy et al., 2021) | S3 | 1 | 27.88 | 67.36 |
| ViT | S3 | 2 | - | - |
| ViT | S3 | 3 | - | - |
| ViT | **S4 (ours)** | 1 | 26.14 | 67.39 |
| ViT | **S4 (ours)** | 2 | 26.19 | 67.55 |
| ViT | **S4 (ours)** | 3 | 27.45 | **67.92** |
| Swin (Liu et al., 2021) | S3 | 1 | 27.58 | 72.4 |
| Swin | S3 | 2 | - | - |
| Swin | S3 | 3 | - | - |
| Swin | **S4 (ours)** | 1 | 25.34 | 72.31 |
| Swin | **S4 (ours)** | 2 | 27.11 | 72.58 |
| Swin | **S4 (ours)** | 3 | 28.43 | **73.09** |
| Shunted (Ren et al., 2021) | S3 | 1 | 21.32 | 75.16 |
| Shunted | S3 | 2 | - | - |
| Shunted | S3 | 3 | - | - |
| Shunted | **S4 (ours)** | 1 | 20.09 | 75.28 |
| Shunted | **S4 (ours)** | 2 | 20.78 | 75.42 |
| Shunted | **S4 (ours)** | 3 | 21.27 | **75.59** |

Table 10: Comparisons of search space exploration of ViT, Swin Transformer, and Shunted Transformer using S3 and **S4**, with backtracking for S3. Results are conducted on Cifar10.

| Model | Exploring approach | backtrack time | Model Constraint(M) | #Params.(M) | Cifar 10 Accuracy |
|---|---|---|---|---|---|
| ViT (Dosovitskiy et al., 2021) | - | - | 30 | 28.71 | 93.59 |
| ViT | S3 | 1 | 30 | 29.32 | 94.23 |
| ViT | **S4 (ours)** | 0 | 30 | 25.39 | **94.32 (+0.09)** |
| Swin (Liu et al., 2021) | - | - | 30 | 27.52 | 94.39 |
| Swin | S3 | 2 | 30 | 29.88 | 95.61 |
| Swin | **S4 (ours)** | 0 | 30 | 28.33 | **95.65 (+0.04)** |
| Shunted (Ren et al., 2021) | - | - | 22 | 21.89 | 97.39 |
| Shunted | S3 | 2 | 22 | 21.28 | 97.6 |
| Shunted | **S4 (ours)** | 0 | 22 | 21.98 | **97.67 (+0.07)** |

Table 11: Comparisons of search space exploration of ViT, Swin Transformer, and Shunted Transformer using S3 and **S4**, with backtracking for S3. Results are conducted on Cifar100.

| Model | Exploring approach | backtrack time | Model Constraint(M) | #Params.(M) | Cifar100 Accuracy |
|---|---|---|---|---|---|
| ViT (Dosovitskiy et al., 2021) | - | - | 30 | 28.89 | 76.45 |
| ViT | S3 | 2 | 30 | 29.1 | 78.02 |
| ViT | **S4 (ours)** | 0 | 30 | 26.45 | **78.68 (+0.66)** |
| Swin (Liu et al., 2021) | - | - | 30 | 27.6 | 79.03 |
| Swin | S3 | 2 | 30 | 28.47 | 79.52 |
| Swin | **S4 (ours)** | 0 | 30 | 26.24 | **80.31 (+0.79)** |
| Shunted (Ren et al., 2021) | - | - | 22 | 21.93 | 84.36 |
| Shunted | S3 | 2 | 22 | 19.4 | 84.78 |
| Shunted | **S4 (ours)** | 0 | 22 | 21.56 | **85.66 (+0.88)** |

# E   EVALUATING PERFORMANCE IMPROVEMENT DURING SEARCH SPACE EVOLUTION

To evaluate the effectiveness of our method, we sample 300 architectures and report the average of their inherited accuracy from supernet in each search iteration. Fig. 5 and Fig. 6 show the results of Swin Transformer and Shunted Transformer on Tiny ImageNet throughout the search space evolution process. In these figures, each data point represents an inherited accuracy of a sampled architecture, while the pink points represent the average results. It can be observed that

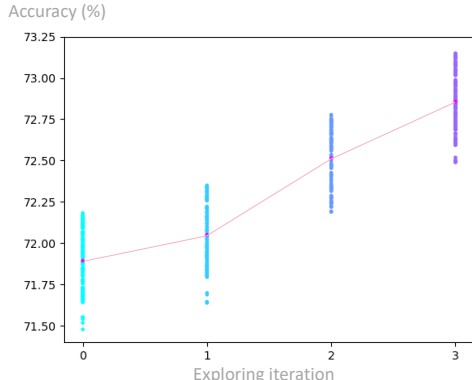

Figure 5: Accuracy on Tiny ImageNet when applying our method to Swin Transformer's search space exploration.

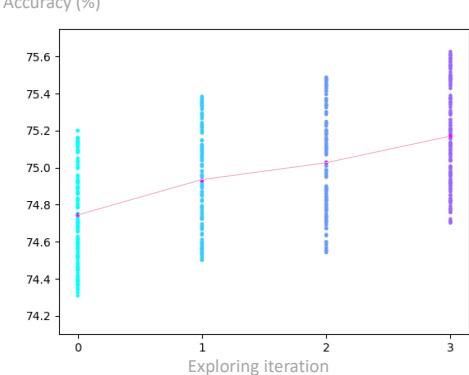

Figure 6: Accuracy on Tiny ImageNet when applying our method to Shunted Transformer's search space exploration.

with an increase in exploration iterations, the performance of the highest and lowest models has improved, leading to an overall upward trend in performance. This once again demonstrates the effectiveness of our method.