# OpenReview forum: "SEEKING THE SEARCH SPACE FOR SIZE-AWARE VISION TRANSFORMER ARCHITECTURE"
_ICLR.cc/2024/Conference — ICLR 2024 Conference Withdrawn Submission_

### Official Review · Reviewer_dqhy · 2023-10-31

**Soundness:** 2 fair
**Presentation:** 2 fair
**Contribution:** 2 fair
**Rating:** 5
**Confidence:** 4

**Summary:**

This paper introduces S4, a novel strategy for exploring the search space in size-aware transformer architecture design. S4 strategically guides the updating process by adhering to predefined size ranges, effectively eliminating the need for unnecessary training iterations and sampling efforts on models that do not conform to size specifications

**Strengths:**

The illustration of using gradients to estimate the search space is exceptionally clear and visually engaging. It effectively aids in understanding a complex concept. The paper's clarity and ease of follow are commendable.

**Weaknesses:**

- **Larger datasets**: The experiments in the paper were carried out on small scale datasets, which may make the results less convincing. It would be better to conduct experiments on larger datasets to validate the results.

- **Variable step size**: The paper claims that fixed step size is better, but it may prohibit the progress of searching for a better search space. It would be better to use variable step size to allow for more exploration of the search space.

- **Clear experimental details**: The experimental details in the paper are not clear. For example, the estimated gradient by 40 sampled subnet is generated by a trained supernet. It would be better to provide more details about how this was done.

- **Ablation study**: The ablation study is missing. As you claimed in the paper, your initial search space is already based on state of the art architectures. I wonder if we started from a bad initial point can still lead us to a better search space? An ablation study would help answer this question.

- **Less human expert experience**: The settings of search space require too much human expert experience. It would be better to automate this process as much as possible.

- **More significant improvements**: The improvements are trivial as shown in table 2. From my understanding, to estimate gradient to guide the search space, you need to train a supernet on current search space, which requires a large amount of computation resources already. However, the final results are almost the same with baseline (do not apply S4), which makes me think the initial search space is already a good choice and there is no need to explore the search space. If there is anything that I misunderstand, please tell me.
I understand. Here are my suggestions:

- **Importance of "Size-aware"**: Limiting the size during the searching process may indeed prohibit further performance improvements. A slightly larger search space that exceeds the required size might be helpful.

- **Code Missing**: Unfortunately, I could not find any information about whether the code will be available.

**Questions:**

see weakness.

---

> ### Author Response · Authors · 2023-11-16
>
> We thanks your valuable comments and feedback. Your raised concerns are addressed in details as follows.
>
> **Weakness part**
> 1. Good suggestion! As S3, the proposed method can also be applied to other datasets on larger scales. However, our computational resources can not afford ImageNet-scale experiment evaluation. For more extensive and fairer comparisons, we conduct the experiments in our paper on larger and more complex datasets than Cifar10 and Cifar100, such as Tiny ImageNet and SUN397, which both contain 100,000 images and 200 and 397 classes, respectively. As shown in Table 3 of the paper, for Tiny ImageNet, our approach achieves 75.59\% classification accuracy in the search spaces of Shunted Transformer and improves 1.52\% in accuracy. Similarly, we achieve 78.19\% on the SUN397 dataset and improve 2.09\% in accuracy.
> 2. That's a very good suggestion. In our current work, the step size design is consistent between different datasets, and all datasets achieve excellent performance with this setting. The manual design effort of this consistent step size is low and the scales of the values are referenced from S3. As long as the step size is small enough, with enough exploration iterations, the search space will finally explore to the best one considering the size constraint. However, we will take your suggestion into consideration for the further improvement of our method.
> 3. Sorry for the insufficient description for this part. Here we provide an example of how we calculate the gradient: Let's say we want to calculate the gradient of the embedding dimension. First we randomly sample a subnet $\alpha$ from the supernet. The depth of $\alpha$ is 12, the embedding dimension is 360, the MLP ratios are all 4 in 12 layers and the number of heads are all 2 in 12 layers. Then we replace the value of the embedding dimension to each of the candidate choices \{336, 360, 384\}. We then get three architectures sharing the same value on all of the search dimensions except for the embedding dimension. We can acquire the inherited accuracy of them from the supernet, for example, 90, 93 and 97. The gradient of the first two architectures will be $(93 - 90) / (360 - 336) = 0.125$. Similar calculation will be conducted in any two of the three architectures. The above procedure will be conducted 40 times and we finally average all the gradients. Same process is applied to all search dimensions. We also provide the detailed algorithm of the whole process in the Supplementary Material.
> 4. That's a good question. Though we currently initial search spaces according to exiting architectures, in Section E in the Supplementary material, it can be observed that the overall performance improves with every iteration. Through our constraint optimization algorithm, the search space will continually explore to a better one. As long as the number of the iterations is sufficient, the initial search space will finally stop at the best space align with the user specified size constraint.
> 5. Thank you for the question. We follow the same setting like S3, which also predefined a range of the search space at the beginning, then apply a search space exploration algorithm to improve them. Compared to S3, we are free from the quantization parameter which needs to be carefully tuned and fully rely on the accuracy gradient ascent algorithm to improve our search space. The manual effort of our method is relatively low.
> 6. We would like to kindly remind you that the comment seems to be incomplete, and we first respond to the part we currently understand of the problem. As you mentioned, we reference the state-of-the-art architectures known for their strong performance to design our initial search space. Instead of manually adjust these architectures, we introduce a constraint optimization algorithm to automate the process. Also, our approach has the plug-and-play characteristic to any initial search spaces including those that are not delicately designed. We believe after sufficient iterations of exploration, we can finally find the best space aligned with the user-specified size constraint, thereby automating and streamlining the overall design process.
> 7. As we mentioned in Section 4 in our main paper, the ultimate goal of our research is to find the model with the best performance given predefined model size. To achieve this, we first acquire the desired model size range from users, and we expect them to give us the maximum model size they can afford considering their resources in the real application scenario. Thus, constraining the model size range in the search space exploration phase can largely reduce the extra effort of training the architectures whose size are beyond the user expected. It is a more efficient way compared to first train a large supernet which includes lots of unqualified architectures and lately discard them until the evolutionary search stage.
> 8.  Good question! We will release our code upon paper acceptance.

---

> > ### Comment · Reviewer_dqhy · 2023-11-20
> > **Remain Rating**
> >
> > Thank you for addressing the concerns and providing detailed responses to the weaknesses identified in your paper. I appreciate the effort you have put into clarifying and improving your work. Based on your responses, I am confident in maintaining the rating of your paper.
> > Also, I agree with Reviewer U6go's second point, there need further theory explanation. I am looking forward to your further improvement

---

### Official Review · Reviewer_kmdm · 2023-10-31

**Soundness:** 2 fair
**Presentation:** 3 good
**Contribution:** 2 fair
**Rating:** 5
**Confidence:** 2

**Summary:**

In this paper, the authors introduce a constrained optimization framework to seek the search space for size-aware transformer architecture. This method combines accuracy gradient ascent with discrete neighboring search space evolution. Extensive experiments have demonstrated the plug-and-play nature and superior performance of this method.

**Strengths:**

1. This paper is written in a clear and accessible manner, making it easy to comprehend.
2. It's great to see that the figures in this paper are simple and reader-friendly, making it accessible to a wider audience.
3. The method presented exhibits a plug-and-play characteristic, thus delivering exceptional performance for downstream tasks.

**Weaknesses:**

1. This paper introduces a search space exploration strategy in a simplistic manner. Considering the extensive literature on search space exploration, this paper lacks innovation.
2. In Table 2, the method is compared only on the Tiny ImageNet dataset, and there is no experimental comparison conducted on the ImageNet dataset. As a result, the genuine efficacy of this approach remains unverified.
3. The predetermined step size of the method may require manual adjustment for different downstream tasks, which, to some extent, still relies on the manual design of the structure.

**Questions:**

Please refer to the weaknesses.

---

> ### Author Response · Authors · 2023-11-16
>
> We thanks your valuable comments and feedback. Your raised concerns are addressed in details as follows.
>
> **Weakness part**
> 1. Thanks for the question. To our best knowledge,  the most prominent work related to the search space exploration of the transformer architecture search is S3. This is the reason why we have conducted extensive experiments and provided analysis and comparison between S3 and our method. We have proposed a constrained optimization framework combining the accuracy gradient ascent with discrete neighboring search space evolution while adhering to strict model size constraints. Our design avoids extra training and sampling efforts on models that do not align with our size expectations and also performs well with adapters for foundation models, which are new features that S3 does not have.
>
> 2. Good suggestion! As S3, the proposed method can also be applied to other datasets on larger scales. However, our computational resources can not afford ImageNet-scale experiment evaluation. For more extensive and fairer comparisons, we conduct the experiments in our paper on larger and more complex datasets than Cifar10 and Cifar100 (both datasets contain images with a resolution of only 32 x 32), such as Tiny ImageNet and SUN397, which both contain 100,000 images and 200 and 397 classes, respectively. As shown in Table 3 of the paper, for Tiny ImageNet, our approach achieves 75.59\% classification accuracy in the search spaces of Shunted Transformer and improves 1.52\% in accuracy. Similarly, we achieve 78.19\% on the SUN397 dataset and improve 2.09\% in accuracy.
>
> 3. Thank you for the question. Our step size design is consistent between different datasets, and all datasets achieve excellent performance with this setting. The manual design effort of a consistent step size is low and the scales of the values are comparable to those of S3. Rather than being a parameter which need an intensive tuning as the parameter in S3, our method are free from the parameters like that and achieve a better performance at the same time.

---

### Official Review · Reviewer_U6go · 2023-10-31

**Soundness:** 2 fair
**Presentation:** 3 good
**Contribution:** 2 fair
**Rating:** 3
**Confidence:** 5

**Summary:**

The paper proposes a constrained optimization framework for transformer architecture search, named S4, that allows the search space to evolve to neighbor search spaces under user-specified constraints, such as model size. The authors demonstrate the effectiveness of S4 through experiments on various benchmarks, including Cifar10, Cifar100, Tiny ImageNet, and SUN397.

**Strengths:**

1. The paper is easy to follow with concise expressions and clear logic.
2. The idea of size-aware search introducing a more effective way to explore architectures is interesting.

**Weaknesses:**

1. The paper does not compare the search costs. This makes it hard to know if the method is efficient.
2. The results reported in the paper are after a retraining process from scratch. I am interested in knowing how the performance of the entire search space's subnet changes with each Evolutionary step of the search space. Specifically, how does the inherited accuracy from the trained supernet change? The paper proposes a rough approximation search algorithm in a highly discretized search space, but such an algorithm does not have a theoretical guarantee. Therefore, rigorous experiments are necessary to validate the effectiveness of the method. Relying solely on final accuracy is not sufficient.
3. Compared to S3, the contribution of this work appears to be insufficient, and the improvements are also small. While the idea of using size-aware techniques to reduce unnecessary searches is interesting, the solutions provided in the paper are somewhat crude. Additionally, the experiments are conducted on small datasets, whereas S3 includes results from mainstream large datasets such as ImageNet, COCO, ADE20K, and VQA2.0. As a result, the experiments in this paper are not so solid or reliable. Since the S3 paper provides ImageNet results, I would like to see S4's performance on ImageNet for a fairer comparison. There is also the possibility of transferring network structures found in small datasets to larger ones.

**Questions:**

After each change in the search space, what is the relationship between the supernet weights at times t and t-1? Are they inherited in a way similar to OnceForAll?

---

> ### Author Response · Authors · 2023-11-16
>
> We thanks your valuable comments and feedback. Your raised concerns are addressed in details as follows.
>
> **Weakness part**
> 1. Thanks for the question. For ViT architecture, the search space exploration takes 12 hours, supernet training takes 3 hours, the evolutionary search takes 0.5 hours, and retraining takes 2.5 hours using a 10-core Intel Xeon CPU with 3GHz and 4 Nvidia Titan V GPUs with 32GB memory on Cifar10 dataset. For the Cifar100 dataset, the search space exploration took 13.5 hours, supernet training took 3.5 hours, evolutionary search took 1 hour, and retraining took 3 hours under the same GPU settings. The search time is similar when we implement the S3 method in the same environment. Though having a similar time cost, we can always find a better architecture than S3 that satisfies the predefined constraint, whereas the quantization hyperparameter in S3 needs to be tuned carefully and requires multiple attempts to make sure to find an architecture aligns with the predefined model size. This proves the efficiency of our method.
>
> 2. That’s a very good observation. We would like to kindly remind you that we have provided the inherited accuracy of 300 sampled architectures during each of the search space exploration iteration in Section E in the Supplementary Material. Fig. 5 and Fig. 6 show the results of Swin Transformer and Shunted Transformer on Tiny ImageNet throughout the search space evolution process. It can be seen that the performance of the highest and lowest architectures has improved, leading to an overall upward trend in performance. This once again demonstrates the effectiveness of our method.
>
> 3. Good suggestion! As S3, the proposed method can also be applied to other datasets on larger scales. However, our computational resources can not afford ImageNet-scale experiment evaluation. For more extensive and fairer comparisons, we conduct the experiments in our paper on larger and more complex datasets than Cifar10 and Cifar100 (both datasets contain images with a resolution of only 32 x 32), such as Tiny ImageNet and SUN397, which both contain 100,000 images and 200 and 397 classes, respectively. As shown in Table 3 of the paper, for Tiny ImageNet, our approach achieves 75.59\% classification accuracy in the search spaces of Shunted Transformer and improves 1.52\% in accuracy. Similarly, we achieve 78.19\% on the SUN397 dataset and improve 2.09\% in accuracy.
>
> **Question part**
>
> Thank you for the question. The answer is yes, we follow the same setting of S3, which trains a once-for-all supernet to obtain a large amount of well-trained subnets.